# Dynamic Changes in Metabolite Accumulation and the Transcriptome during Leaf Growth and Development in *Eucommia ulmoides*

**DOI:** 10.3390/ijms20164030

**Published:** 2019-08-18

**Authors:** Long Li, Minhao Liu, Kan Shi, Zhijing Yu, Ying Zhou, Ruishen Fan, Qianqian Shi

**Affiliations:** 1Northwest Agriculture and Forestry University, College of Forestry, Taicheng Road No. 3, Yangling 712100, Shaanxi, China; 2Northwest Agriculture and Forestry University, College of Enology, Taicheng Road No. 3, Yangling 712100, Shaanxi, China; 3Northwest Agriculture and Forestry University, College of Landscape Architecture and Arts, Taicheng Road No. 3, Yangling 712100, Shaanxi, China

**Keywords:** *Eucommia ulmoides*, metabolome, flavonoid, MYB (v-myb avian myeloblastosis viral oncogene homolog)

## Abstract

*Eucommia ulmoides* Oliver is widely distributed in China. This species has been used mainly in medicine due to the high concentration of chlorogenic acid (CGA), flavonoids, lignans, and other compounds in the leaves and barks. However, the categories of metabolites, dynamic changes in metabolite accumulation and overall molecular mechanisms involved in metabolite biosynthesis during *E. ulmoides* leaf growth and development remain unknown. Here, a total of 515 analytes, including 127 flavonoids, 46 organic acids, 44 amino acid derivatives, 9 phenolamides, and 16 vitamins, were identified from four *E. ulmoides* samples using ultraperformance liquid chromatography–mass spectrometry (UPLC-MS) (for widely targeted metabolites). The accumulation of most flavonoids peaked in growing leaves, followed by old leaves. UPLC-MS analysis indicated that CGA accumulation increased steadily to a high concentration during leaf growth and development, and rutin showed a high accumulation level in leaf buds and growing leaves. Based on single-molecule long-read sequencing technology, 69,020 transcripts and 2880 novel loci were identified in *E. ulmoides*. Expression analysis indicated that isoforms in the flavonoid biosynthetic pathway and flavonoid metabolic pathway were highly expressed in growing leaves and old leaves. Co-expression network analysis suggested a potential direct link between the flavonoid and phenylpropanoid biosynthetic pathways via the regulation of transcription factors, including MYB (v-myb avian myeloblastosis viral oncogene homolog) and bHLH (basic/helix-loop-helix). Our study predicts dynamic metabolic models during leaf growth and development and will support further molecular biological studies of metabolite biosynthesis in *E. ulmoides.* In addition, our results significantly improve the annotation of the *E. ulmoides* genome.

## 1. Introduction

*Eucommia ulmoides* Oliver is the only species in the family Eucommiaceae and is widely distributed in China [1,2]. *E. ulmoides* has long been used in medicine due to the high concentrations of chlorogenic acid (CGA), rutin, quercetin, iridoid, and α-linolenic acid in its leaves and bark [3]. The pharmacology and efficacy of *E. ulmoides* have been well documented in ancient Chinese medicinal books such as Shennong’s *Classic of Materia Medica* and *Compendium of Materia Medica*. In addition, this species is also an important source of natural rubber [4,5]. Furthermore, *E. ulmoides* also has important applications in landscaping and soil and water conservation [3]. Due to its many economic, ecological and social benefits, *E. ulmoides* is one of the most important forest resources in China.

*E. ulmoides* has many beneficial effects on human health, such as neuroprotection [6]; bone loss prevention [7]; learning and memory improvement [8]; insulin resistance amelioration [9]; antibacterial [10], lipid-lowering and anti-obesity effects [11]; and osteoarthritis treatment effects [12]. Moreover, 204 compounds have been isolated from and identified in *E. ulmoides*, including lignans, iridoids, flavonoids, phenols, steroids, and terpenes [13]. Flavonoids are a large class of secondary metabolites and are widely distributed in plants. Flavonoids can be divided into seven subclasses: flavones, flavonols, flavandiols, chalcones, anthocyanins, condensed tannins and aurones [14]. Flavonoids were the first pigments characterized as defence compounds and signalling molecules that can withstand a wide range of biotic and abiotic stressors in plants, and flavonoids can prevent or defend against diseases in humans [15]. So far, the genes involved in flavonoid biosynthesis have been well characterized in many model plants. Many functional genes including *chalcone synthase* (*CHS*), *flavanone 3-hydroxylase* (*F3H*), *flavonoid 3′-hydroxylase* (*F3′H*), *chalcone isomerase* (*CHI*), *flavonol synthase1* (*FLS1*), *dihydroflavonol 4-reductase* (*DFR*), and *leucoanthocyanidin dioxygenase* (*LDOX*) play critical roles in the production of flavonoids [16,17]. In *Arabidopsis*, three MYB members (MYB11, MYB12, and MYB111) regulating the expression of several ‘early’ flavonoid biosynthetic genes, and PAP1, PAP2, GL3, TT8 and TTG1, which are components of the MYB/bHLH/WDR (MBW) transcriptional complexes mediate the ‘late’ anthocyanin biosynthesis genes [18,19]. A good deal of evidence suggested that hormone signalling pathways have been shown to be involved in these regulations network. The accumulation of flavonoids induced by abscisic acid (ABA), jasmonate (JA), and cytokinins, while repressed by gibberellic acid (GA), ethylene, or brassinosteroids (BRs). Although the biosynthesis of flavonoid is relatively conserved across different organisms, the overall molecular mechanisms, the prediction of hub genes, and the accumulation level of flavonoids during leaf growth and development were still not reported in *E. ulmoides* [20,21].

CGA (chlorogenic acid) is one of the most interesting active herbal ingredients in *E. ulmoides* and belongs to the phenylpropanoid family of compounds. This compound can reduce blood pressure and has been investigated for possible anti-inflammatory effects. CGA exhibits anti-obesity effects in mice and can improve lipid profiles and decrease the obesity-associated hormone ratio [22,23]. Previous studies have suggested that CGA is synthesized in various plant species, including apple, artichoke, coffee, eggplant, tobacco, tomato, pear and *Lonicera macranthoides* [24,25,26,27], and high *HQT* expression appears to be closely correlated with CGA accumulation. Overexpression of *AtPAL2* in tobacco resulted in a twofold increase in CGA, while silencing of *HQT* in *AtPAL2-*overexpressing plants resulted in a 50% decrease in CGA [28]. To date, few studies have utilized omics technology to identify the type and variety of metabolite effects in *E. ulmoides*, despite many phytochemical studies that have focused on several important herbal ingredients. The dynamic changes in the accumulation of various metabolites during *E. ulmoides* leaf growth and the overall molecular mechanisms involved in metabolite biosynthesis remain unknown.

Recently, bioinformatics studies of *E. ulmoides* were conducted to better understand its growth and physiological characteristics. Transcriptome sequencing of *E. ulmoides* seeds identified many key regulators involved in fatty acid biosynthesis, including 3-oxoacyl-ACP reductase, β-hydroxyacyl-ACP dehydratase, and β-ketoacyl-ACP [29]. A total of 24 families, including those encoding enolase, fructose-bisphosphate aldolase, and triosephosphate isomerase, play important roles in developing *E. ulmoides* seeds [30]. The identification of *Eucommia* rubber (EU-rubber) biosynthesis-associated miRNAs in the leaf and fruit indicated that *Eu-miR14*, *Eu-miR91*, *miR162a*, *miR166a*, *miR172c*, and *miR396a,* as well as the associated target genes, such as *HD-ZIP*, *AP2-EREBP*, and *GRF*, serve as potential regulators in EU-rubber accumulation [31]. Transcriptome sequencing and single nucleotide polymorphism (SNP)-calling analyses confirmed that the APETALA3-like gene was most likely involved in sex determination in *E. ulmoides* [31]. Single-molecule sequencing technology has significantly improved read lengths compared with second-generation sequencing (SGS) technologies [32] and avoids the need for transcriptome assembly, which is essential for SGS technologies. This method can significantly improve gene models and identify alternative splicing (AS) events. Recently, single-molecule sequencing technology was used to characterize transcriptomic complexity in many species, such as sorghum, maize, cotton, moso bamboo and *Populus* [33,34,35,36,37]. In contrast, the transcriptome diversity based on single-molecule sequencing technology remains largely unknown in *E. ulmoides*.

In the present study, metabolome profiling was performed to identify categories of metabolites and their dynamic changes during leaf growth and development. Single-molecule sequencing data were used to ensure wide coverage of transcript isoforms in *E. ulmoides.* To compare expression levels between different samples and explore potential factors involved in metabolite biosynthesis and leaf growth, SGS was carried out using isoform models identified by single-molecule sequencing. Our findings provide new insights into the molecular mechanisms associated with the biosynthesis and regulation of flavonoids and phenylpropanoids during *E. ulmoides* leaf growth and development and highlight the usefulness of an integrated approach for understanding this process. Besides, our single-molecule sequencing results show that characterization of the *E. ulmoides* genome database is far from complete, and our results significantly improve the annotation of *E. ulmoides* genome.

## 2. Results

### 2.1. Anatomical and Physiological Analysis of *E. ulmoides* Leaves

*E. ulmoides* leaves developed from the leaf primordium, which is located around the growing point of the leaf bud (Figure 1B). In the initial growth stage, both the upper and lower epidermises of the *E. ulmoides* leaves were covered with trichomes. As the leaves continued to grow, the trichomes gradually fell off. The stomata were distributed on the lower epidermis of the leaves (Figure 1C). The cross-sectional area of the main vein in the middle part of leaves reached 7.07 mm^2^ and stopped increasing when the leaf length reached 3 cm (Figure 1B). The leaf thickness gradually increased from L3 (138.5 ± 12.1 μm) to L4 (172.4 ± 15.3 μm), although the leaf area no longer increased.

### 2.2. Overview of Single-Molecule Long-Read Sequencing and SGS Results

To obtain comprehensive annotations of *E. ulmoides* genes and identify the key molecular basis of physiological processes involved in *E. ulmoides* growth and development, we performed single-molecule long-read sequencing (SLS) using PacBio RS II and SGS with the Illumina platform. In total, three SMRT (Single Molecule, Real-Time Sequencing) cells generated 356,201 ROIs (Reads of insert) (Appendix A and Appendix A). Among these ROIs, 277,526 sequences with the entire transcript region from the 5′ end to the 3′ end and containing the poly (A) tail were identified as FLNC (Full length readsnon-chimeric) reads (Appendix A). We used ICE (Iterative isoform-clustering) [38], an isoform-level clustering algorithm, to improve consensus accuracy and cleaned FL (Full length) consensus sequences from ICE using SMRT Analysis (v2.3.0) software. A total of 143,375 consensus isoforms were obtained, 101,427 of which were high-quality consensus transcript sequences (Appendix A). To avoid and correct the high error rates compared with those of the Illumina platform [33], we generated 266,404,623 PE (paired-end) reads using the Illumina platform to correct the single-molecule long reads (Appendix A). Finally, 69,020 transcripts corresponding to 25,200 gene loci were obtained after elimination of redundancy by Cogent software, including 56 WRKYs, 303 MYBs, 16 bHLH, 5 GRF (Growth-regulating factor), 13 E2F, et al. (Appendix A). Among these, 2880 novel loci were not annotated in *E. ulmoides* genome database [2] (Appendix A). A total of 14,149 genes produced only a single transcript, while the other 11,051 genes produced two or more transcripts, indicating that these genes underwent AS (Figure 2A). Based on ASTALAVISTA (Alternative Splicing landscape), the AS (Alternative splicing) events were classified into five main categories: intron retention, skipped exon, alternative 5′ site (5′ SS), alternative 3′ site (3′ SS) and mutually exclusive exon. Among all AS events, intron retention was the dominant form, accounting for 54.72% of all the alternatively spliced transcripts (Figure 2b and Appendix A).

A total of 45,941 open reading frames (ORFs) were predicted by TransDecoder, 36,581 of which were high-integrity ORFs with intact coding sequences (CDSs) from initiation codons to termination codons. We randomly selected four genes including five AS events to validate the accuracy of AS events using reverse transcription polymerase chain reaction PCR (RT-PCR). As shown by a gel banding pattern, the size of each amplified fragment was consistent with that of predicted fragment (Figure 2C). Conserved domain analysis revealed that many AS events exerted on domain modification and lost, which may lead to functional changes. The *EUC14737-RA*, a *Camellia sinensis* mitoferrin-like gene could generate two splicing isoforms *EUC14737-RA* and *PB.1012.2* due to intron retention. Amino acid sequence prediction revealed *PB.1012.2* contained an incomplete mitochondrial carrier domain. However, some AS events only influenced sequence length of amino acid, and the conserved domains did not change, such as *EUC12362-RA* (Figure 2C).

Four computational methods were used to identify lncRNAs from the 69,020 PacBio Iso-Seq isoforms, and a total of 2075 lncRNAs were identified (Appendix A). Of the 25,200 genes detected by Iso-Seq, 7064 genes have at least one poly(A) site, and 303 genes have more than six poly (A) sites (Figure 2D). In addition, we observed clear nucleotide bias around poly(A) sites in *E. ulmoides* with an enrichment of uracil upstream and adenine downstream of the cleavage site in 3′ UTRs (Figure 2E). A total of 2715 full-length transcripts that could be mapped to two or more loci in the genome were considered fusion transcripts (Appendix A). The PCCs (Pearson correlation coefficient) of different samples based on isoform expression indicated that two adjacent stages showed much larger correlation coefficients than two non-adjacent stages (Appendix A). To validate the reliability of the expression data obtained by RNA-seq, qRT-PCR was performed on 10 leaf development-, flavonoid- and EU-rubber biosynthesis-related isoforms. As expected, the qRT-PCR data were consistent with the RNA-seq data, with PCCs ranging from 0.918 to 0.989 (Appendix A).

### 2.3. Different Regulatory Mechanisms Involved in Different *E. ulmoides* Growth Stages

To identify key factors involved in the different *E. ulmoides* growth stages, weighted correlation network analysis (WGCNA) and GO enrichment analysis were conducted (Figure 3). Eleven major co-expression isoform modules were detected among the 12 transcription samples (Figure 3A,B). The flavonoid biosynthetic process, the flavonoid metabolic process and many photosynthesis- and plastid-related biological processes, such as photosystem II assembly, chloroplast organization, carotenoid metabolic process, and chloroplast organization, were highly enriched in the blue module, corresponding to the isoform that showed gradually increasing expression during leaf growth and development. Several photosynthesis-related biological processes and many metabolic processes were simultaneously enriched in the dark orange (L2) and purple (L3) modules. The isoforms involved in fatty acid-, ethylene-, water response- and pentacyclic triterpenoid compound-related biological processes were significantly enriched in the brown module, corresponding to isoforms showing high expression levels in L1. DNA replication and several cell cycle-related processes were significantly enriched in the light cyan and light green modules, indicating that these isoforms were highly expressed in L1 and L2. A majority of cell growth-associated isoforms as well as many flavonoids and anthocyanin biosynthesis related isoforms were significantly enriched in light cyan, indicating these isoforms were highly expressed in L2 (Figure 3C).

Hormone metabolism and signalling, aquaporins, the cell cycle and Transcription factors play important roles in leaf growth, and hormone metabolism and signalling and TFs play important roles in phenylpropanoid biosynthesis and flavonoid biosynthesis. Thus, we analysed the expression patterns of these isoforms. The expression of most *AP2*, *auxin response factor (ARF)*, *ZF-HD* and *bZIP* isoforms peaked in L1 or L2. The expression of most MYB isoforms was high in L2 or L4. The two identified *E2Fs* were predominantly expressed in L2 (Figure 3D and Appendix AA). Cell cycle-related isoforms were highly expressed in L2 (Appendix AB). Among aquaporins, tonoplast intrinsic protein (TIP) and plasma membrane intrinsic protein (PIP) were highly expressed in L1 and L2, while TIP and small and basic intrinsic protein (SIP) were highly expressed in Appendix A (Appendix AC). Most auxin and gibberellin signalling-related isoforms were highly expressed in L1 or L2, such as *GID1* and *GH3* in L1 and *ARF*, *AUX1/LAX*, and *DELLA* in L2. Of the isoforms related to jasmonic acid (JA) signalling, *JAZ* and *MYC2* showed low expression levels in L4, whereas *JAR1* and *COI1* showed high expression levels in L4 (Appendix AD).

### 2.4. Metabolomic Analysis

A total of 515 metabolites, including 127 flavonoids, 46 organic acids, 44 amino acid derivatives, 8 isoflavones, 9 phenolamides, and 16 vitamins, were identified from four *E. ulmoides* leaf samples using ultraperformance liquid chromatography-mass spectrometry (UPLC-MS; Appendix A). The PCCs of different samples based on metabolite levels indicated that, similar to the PCCs based on isoform expression levels, two adjacent stages showed much stronger correlations than two non-adjacent stages. Moreover, L1 showed a close relationship with L2, and L3 showed a close relationship with L4 (Figure 4A). The heat map generated based on metabolite levels indicated that more metabolites showed high concentration levels in L1 and L2 than in L3 and L4 (Figure 4B). The number of differentially accumulated metabolites ranged from 84 to 142. The largest difference was between L2 and L3, including 95 metabolites with high concentrations in L2 and 47 metabolites with high concentrations in L3 (Figure 4C).

To comprehensively analyse the changes in metabolite concentrations during leaf growth and development, WGCNA (weighted correlation network analysis) was used to extract the dynamically variable modules of metabolites. Five modules of strongly correlated metabolites (blue, brown, turquoise, yellow and grey) were identified during different stages of metabolite development (Figure 5A,B). The concentrations of most anthocyanins, amino acids, amino acid derivatives, lipid fatty acids, nucleotides and their derivatives and hydroxycinnamoyl derivatives peaked in L2. The abundances of most catechin derivatives, lipid glycerolipids, and lipid glycerophospholipids peaked in L3. Cholines, coumarins and proanthocyanidins were present at high concentrations in L4. The concentrations of most flavone C-glycosides, flavanones, and flavonols and quinate and its derivatives peaked in L2 or L4 (Figure 5C).

Metabolomic analysis showed that most of flavonoids were highly accumulated in growing leaves, followed by old leaves. In order to assess the dynamic change of total flavonoid content during leaf growth and development, we used spectrophotometric method. Similar with the metabolomics analysis results, the concentration of flavonoid content increased first, and then decreased sharply to 3.2% at S7, corresponding to L3 stage which used for RNA-seq and metabolome profiling (Appendix AA). Finally, the flavonoids content increased again. HPLC was carried out to comprehensively analyse the dynamic changes in the levels of two important herbal ingredients, namely, CGA and rutin. The temporal changes in CGA concentration present a unimodal pattern in nature. The peak appeared at S9, corresponding to the previous stage of L4, which was used for RNA-seq (Appendix AB). The rutin content was maintained at a high level at first and gradually decreased thereafter (Appendix AC).

### 2.5. Analysis of Isoforms and Metabolites Associated with Phenylpropanoid and Flavonoid Biosynthesis

Previous studies suggested that CGA and rutin are important active herbal ingredients in *E. ulmoides* [6]. Thus, we investigated the branch of the phenylpropanoid and flavonoid biosynthesis pathways leading to CGA and rutin biosynthesis. Figure 6 shows that the compositions of compounds in the phenylpropanoid, flavone and flavonol biosynthesis pathways were significantly different depending on growth stage. Phenylalanine ammonia-lyase (PAL) is the first key enzyme of the phenylpropanoid pathway, and most *PAL* members showed relatively balanced expression levels in the first three growth stages but showed low expression in L4. These proteins regulate the synthesis of cinnamic acid and always show low accumulation levels in L4. The expression levels of most *F3′H*, *FLS*, *CHI* and *CHS* isoforms were relatively high in L2, resulting in high concentrations of naringenin chalcone, aromadendrin, kaempferol and quercetin in the L2 stage. p-Coumaric acid was highly accumulated in L1 under the regulation of *C4M*, which showed high abundance in L1. Rutin showed gradually decreasing accumulation under the control of *FG*. The expression of two *HQT* isoforms was relative equilibrium in four samples. The *HCT* showed a high expression level in L2, followed by L1. Under the control of *HQT* and *HCT*, the CGA showed a low concentration in L1 and L2.

To understand the regulatory network of phenylalanine- and flavonoid-related metabolites implicated in the leaf growth and development, we carried out correlation tests between quantitative changes of metabolites and transcripts. A total of 402 isoforms had a strong correlation (R ≥ 0.95 or R ≤ −0.95) with 77 phenylpropanoid- or flavonoid-related metabolites (Figure 7). The networks showed that the transcripts and metabolites were grouped into four major clusters (1, 2, 3 and 4). The maximum numbers of positive correlations of isoforms in cluster 1, cluster 2 and cluster 3 were *EUC03052-RA* (MYB), *EUC25953-RA* (WD40) and *EUC23044-RA* (bHLH), respectively. Based on the predicted interaction network, the main hub isoform *EUC03052-RA* positively correlated with seven *WRKY*, three *PP2C*, three *peroxidase*, two *NAC*, two *bZIP*, two *GID1* and one each of the *PAL*, *NPR1*, *CAD*, *GH3*, but negatively correlated with one *WD40*. In addition, the expression of *EUC03052-RA* was positively correlated with the levels of p-coumaric acid, sinapyl alcohol, ferulic acid and coniferyl alcohol. The concentration of rutin was positively correlated with two isoforms, namely, *PB.2810.2* (*MYC2*) and *PB.13759.1* (*MYB*), but negatively correlated with three *gluA*, two *MYB*, two *NAC,* two *DFRA* and one each of the *UDPG, DFRA, bglA, WRKY, TGA, BKI1, WD40.* The CGA content was negatively correlated with *PB.9275.1* (*MYB*), *PB.15156.1* (*AUX1/LAX*), *PB.14036.3* (*AUX/IAA*), *EUC24176-RA* (*SAUR*), *EUC23783-RA* (*MYB*), *EUC19303-RA* (*SAUR*) and *EUC15659-RA* (*CYCD3*).

## 3. Discussion

### 3.1. The Landscape of *E. ulmoides* Transcript Diversity

To date, most studies in *E. ulmoides* have been based mainly on gene expression analysis by SGS technologies, with the aim of identifying key factors involved in the rubber synthesis and organ development of *E. ulmoides* [39,40]. Characterization of AS will facilitate functional genomics in *E. ulmoides*. Without accurate isoform annotation, studies cannot determine functional differences in isoforms. Single-molecule technology eliminates the need for assembly. This technology has led to the discovery of thousands of novel gene loci and alternatively spliced isoforms in many species, such as sorghum, maize, cotton and *Populus* [33,34,35,36,37]. However, this technology has not been applied in *E. ulmoides.* Here, a total of 25,200 expressed genes including 2880 novel locus genes were identified in *E. ulmoides* leaves based on SLS technologies, and 40.9% of the *E. ulmoides* genes underwent AS events. Similar to other reported plants [33,34,35,36,37], intron retention represents the largest proportion of AS and the number of genes with intron retention events was estimated to be 54.72% of all genes undergoing AS in *E. ulmoides*.

### 3.2. The Molecular Mechanism Involved in Flavonoid and Phenylpropanoid Synthesis During *E. ulmoides* Growth and Development

Flavonoids are a large group of polyphenolic compounds and a structurally diverse class of plant secondary metabolites [41,42]. A total of 127 flavonoids were identified in *E. ulmoides* leaves, most of which were highly accumulated in growing leaves, followed by old leaves (Figure 5C and Appendix A). Based on previous studies [15], the high abundance of flavonoids might help growing leaves to confer protection against various abiotic and biotic stresses. Consistent with the dynamic changes in flavonoid accumulation, expression analysis of many key flavonoid biosynthetic isoforms, such as *CHS, DFR,* and *ANS*, revealed high expression levels in these two samples (Figure 6). PAL is one of the key enzyme families involved in anthocyanin, flavonoid and phenylpropanoid biosynthesis [22,42]. Expression analysis showed that most *PAL*s exhibited low abundances in old leaves, associated with low accumulation of the product cinnamic acid in old leaves. R2R3-MYB TFs regulate flavonoid and phenylpropanoid accumulation by regulating the target isoforms *CHS*, *CHI*, and *DFR* [42]. In the co-expression network, several *MYB* and *bZIP* isoforms were considered ‘hubs’ because of their high connectivity (Figure 7). The *MYB* gene family is one of the largest and most functionally diverse families of TFs in the plant kingdom [43], and increasing evidence suggests that flavonoid and phenylpropanoid biosynthesis is regulated by individual MYBs or by the MBW complex formed by MYB, bHLH and WD40 [44,45]. In the subnet, *EUC03913-RA* (*MYB*) had 21 edges and positively correlated with the expression of *EUC00036-RA* (*bHLH*), *EUC24478-RA* (*NAC*), *EUC17504-RA* (*CYCD3*), *EUC14021-RA* (*UG*), *EUC06366-RA* (*JAZ*), *EUC08247-RA* (*MYB*), *EUC16769-RA* (*MYB*), *EUC17607-RA* (*BZR1/2*), *PB.5344.2* (*peroxidase*), *PB.6094.2* (*JAZ*) and *PB.7995.1* (*BSK*), while negatively correlated with the expression of *EUC23044-RA* (*bHLH*) and *PB.3806.1* (*peroxidase*) (Figure 7). There were also different degrees of correlation (both positive and negative) between the expression levels of the *MYBs* and the accumulation of metabolites determined by UPLC-MS. For example, the hub isoform *EUC16769-RA* (*MYB*) showed a negative correlation with the accumulation of naringenin C-hexoside, naringenin 7-O-glucoside (prunin) and myricetin 3-O-galactoside during leaf growth and development (Figure 7), In *Populus*, MYB182 acts as a repressor that reduces the abundance of flavonoids, especially proanthocyanidins and anthocyanin, by regulating flavonoid-related gene expression [46]. A similar phenomenon has consistently been observed in grapes [47], indicating that many MYBs act not only as positive regulators but also as negative regulators in the flavonoid biosynthesis pathway. In the co-expression network, rutin accumulation presented a negative correlation with the expression of two *MYB* isoforms (*PB.13750.1* and *EUC00390-RA*), and low accumulation of rutin was observed in old leaves (Figure 7). FtMYB13, FtMYB14, FtMYB15, and FtMYB16, which are in the jasmonate-responsive subgroup, repress rutin biosynthesis by regulating *PAL* gene expression in *Fagopyrum tataricum* [48]. Thus, we deduced that a high abundance of *EUC00390-RA* and *PB.13750.1* in old leaves significantly repressed *PAL* expression, resulting in low accumulation of rutin in old leaves, and that this process was regulated by JA signalling [49].

High accumulation of flavonoids, important defence compounds, in plants can enhance stress tolerance by inhibiting the generation of reactive oxygen species (ROS) via JA signalling regulation [49]. In this study, we used two types of fully expanded leaf samples, namely, new leaves and old leaves, to compare the differences in metabolite accumulation and identify potential molecular mechanisms involved in metabolite biosynthesis. UPLC-MS analysis indicated that a majority of the flavonoids exhibited higher accumulation in old leaves than in young leaves (Figure 5C and Appendix A). *JAZs,* which act as negative regulators of JA, exhibited low expression levels in old leaves compared with young leaves. In *Arabidopsis*, JA activates the degradation of *JAZs* in an SCF^COl1^ complex-dependent manner to disrupt the interaction between JAZs and bHLH/R2R3–MYB complexes and to activate WD40/bHLH/R2R3-MYB complexes. Thus, low abundances of *JAZs* in old leaves lead to stimulation of WD40/bHLH/R2R3-MYB complexes, activating the expression of flavonoid biosynthesis-associated isoforms and in turn leading to high accumulation of flavonoids in old leaves [50,51]. DELLA, a plant growth inhibitor [52,53,54], showed high abundance in growing leaves (Appendix AD). Much evidence suggests that DELLA proteins positively regulate nitrogen deficiency-induced anthocyanin accumulation through direct interaction with PAP1 [55]. The high abundance in growing leaves might be essential for anthocyanin biosynthesis, resulting in a much higher concentration of anthocyanins in growing leaves (Figure 5C).

UPLC-MS analysis indicated that CGA accumulated steadily to a high concentration during leaf growth and development (Figure 6A). The expression of *PAL* and *HQT* is correlated with CGA concentration [20]. Mining of the single-molecule sequencing data revealed two *HQT* isoforms in the *E. ulmoides* genome. In potato, HQT can catalyse a reverse reaction in which caffeoyl-CoA is synthesized from CGA [20,21]. Thus, the high expression level of *PB.12127.1* (HQT) in L2 resulted in a low concentration of CGA in L2.

### 3.3. The Molecular Mechanism Involved in *E. ulmoides* Growth and Development

Auxin is involved in the rapid and specific regulation of auxin-inducible genes at the transcriptional level. Increased *AUX/LAX1* activity reinforces the auxin-dependent induction of certain cell-wall-remodelling enzymes (Appendix AD), which can promote cell separation [56]. High accumulation of auxin signalling related genes in growing leaves is essential for auxin polar transport and cell division. However, many new evidences suggested that auxin regulates flavonoids especially anthocyanin biosynthesis depend on the Aux/IAA–ARF signaling pathway [49,57]. In cluster 2 of co-expression network, we observed a positive correlation between the levels of auxin signaling genes and anthocyanins. For example, the accumulation of delphinidin showed positive correlation with *EUC24176-RA* (*SAUR*), *PB.15156.1* (*AUX1/LAX*) and *EUC07748-RA* (*ARF*) expression, and the accumulation of cyanidin O-acetylhexoside showed positive correlation with *PB.15156.1* (*AUX1/LAX*), *PB.14036.3* (*AUX/IAA*), *EUC24176-RA* (*SAUR*), *EUC07748-RA* (*ARF*) and *EUC07451-RA* (*AUX/IAA*) expression. These results suggested the involvement of auxin in anthocyanin biosynthesis. Aquaporin participates in water absorption and cell elongation during auxin-induced growth [58]. Thus, a high abundance of *aquaporins* is essential for cell differentiation in growing leaves (Appendix AC). Cell cycle- and cell division-associated isoforms, such as *cyclin A*, *B* and *D*, as well as many plant growth- and development-related TFs, such as *AP2* and *E2F*, were highly expressed in the growing leaves (Appendix AD), which affected their cell division, as captured by microscopy. In addition, *aquaporins* and cell cycle-, plant growth- and development-related TFs exhibited much higher expression levels in young leaves than in old leaves. Although the leaf area no longer increased when the young leaves were fully expanded, the high abundances of these genes resulted in an increase in leaf thickness (Figure 1B).

## 4. Materials and Methods

### 4.1. Plant Materials

Grafted seedlings of “Qinzhong No. 1” were planted in the field in March 2016 at the nursery of Northwest A&F University, Yangling, Shaanxi. One-year-old *E. ulmoides* plants with similar growth patterns (seedling height: 1 m, ground diameter: 8 mm) were used as rootstocks. Two-year-old saplings were used for sample collections. Based on growth stages, we selected four representative tissues (Figure 1A), namely, leaf buds (rudimentary stems with many spires, L1), growing leaves (3 cm long, L2), young leaves (new, fully expanded leaves, L3) and old leaves (60 days after full expansion, L4), for further transcriptomic and metabolomic analysis. For each transcriptome and metabolome sample, ten leaf buds or leaves from one individual and tissues collected from quintuplicate independent individuals constituted one biological replicate, and three biological replicates were examined for each sample.

### 4.2. Paraffin Sectioning

The leaf and leaf bud samples at different growth stages were fixed in formalin acetic acid (FAA), dehydrated in an ethanol series, infiltrated with xylene, and further embedded in paraffin. The embedded tissue samples were then sectioned at an 8-μm thickness and stained with safranin and fast green. Images were captured with an Olympus BX-51 microscope (Olympus, New York, NY, USA) with a digital image acquisition system.

### 4.3. SEM (Scanning Electron Microscope)

The epidermis of leaves and leaf buds was observed by scanning electron microscopy (SEM). After 4% glutaraldehyde fixation, all the tissues were washed with 0.1 M phosphate-buffered saline (PBS) three times. Then, the samples were dehydrated with an ethanol series, and the ethanol was replaced with iso-amyl acetate three times. Then, all the samples were critical point dried using a CO_2_ critical point dryer (EMS850, Quorum, East Sussex, UK) and covered with gold-palladium using an ion coating machine (EIKO IB-3, EIKO, Tokyo, Japan). Finally, images were captured by a scanning electron microscope (EDAX JSM-6360LV, Jeol, Tokyo, Japan) using a 2-kV accelerating voltage.

### 4.4. RNA Extraction and Quality Assessment

Total RNA was extracted using a Plant RNA Extraction Kit (Omega, Omega Bio-Tek, Shanghai, China) and treated with RNase-free DNase I to remove residual DNA contamination. Then, the purity and integrity of the RNA were checked using 1.2% agarose gels, a NanoPhotometer spectrophotometer (N50, Implen, München, Germany), and an Agilent Bioanalyzer 2100 system (Agilent Technologies, Santa Clara, CA, USA).

### 4.5. PacBio Iso-Seq Library Preparation, Sequencing and Analysis

One Pacific Biosciences Isoform Sequencing (PacBio Iso-Seq) library was prepared by combining the total RNA from L1, L2, L3 and L4 in equal amounts according to the PacBio protocol. The cDNA products were used to generate a SMRTbell template library according to the vendor’s instructions. The library was prepared for sequencing by annealing a sequencing primer and adding polymerase to the primer-annealed template. The polymerase-bound template was bound to MagBeads, and sequencing was performed on a PacBio RS II instrument (Pacific Biosciences, California, CA, USA) with three single-molecule, real-time (SMRT) cells.

Analysis of the Iso-Seq data was performed by using SMRT Analysis software v3.0. First, the full-length non-chimeric (FLNC), non-FL and chimeric reads of inserts (ROIs) were separated by the pbtranscript.py script. The FLNC ROIs were identified based on their 5′ adapters, 3′ adapters and poly (A) tail. Next, the poly (A) tails and adapter sequences were removed. The FLNC ROIs were clustered by iterative clustering for error correction (ICE) using SMRT Analysis (v2.3.0) software (PacBio, Menlo Park, CA, USA) to generate the consensus clusters of all FLNC, non-FL, and chimeric sequences. This error self-correction was executed by a quality-aware algorithm in Quiver software to finally obtain the clean FL consensus sequences. The non-redundant cleaned dataset consisted of high-quality consensus transcripts (expected accuracy ≥99% or quality value (QV) ≥ 30) and low-quality transcripts, which were obtained due to insufficient coverage or derived from rare transcripts.

AS event identification was performed by ASTALAVISTA and classified as described previously [59]. Transcript isoforms were translated based on the longest ORF by Getorf. Long non-coding RNA (lncRNA) identification and fusion transcript identification were executed as described in previous studies [60,61].

### 4.6. Illumina Transcriptome Library Preparation, Sequencing and Analysis

Illumina transcriptome sequencing was performed to avoid and correct the high error rates of PacBio Iso-Seq [33], and further used to explore potential factors involved in metabolite biosynthesis and leaf growth. A total of 1 μg of RNA per sample was used as input material to generate sequencing libraries with an NEBNext ^®^ Ultra^TM^ II RNA Library Preparation Kit for Illumina (**NEB #**E7775, NEB, Ipswich, England) according to the vendor’s instructions. The libraries were sequenced on an Illumina HiSeq X Ten platform and paired-end (PE) reads were generated (read length = 150 bp*2). For each sample, three biological replicates were used independently for library construction.

Raw RNA-seq reads generated from the Illumina sequencing platform were filtered using in-house Perl scripts with the default parameters to remove low-quality reads, including reads containing adapters, reads containing poly-N sequences and reads of low quality. These clean reads were then mapped to the reference genome sequence using Tophat2 [2]. Only reads with a perfect match or one mismatch were further analyzed and annotated based on the reference genome. Isoform expression levels were calculated by FPKM. Differential expression analysis between two conditions/groups was performed using the DESeq R package (1.10.1) [62]. An isoform was considered differentially expressed if it exhibited a fold change > 2 with an adjusted e-value (false discovery rate, FDR) < 0.05, as determined by the Benjamini and Hochberg method. The WGCNA (weighted correlation network analysis) (v1.42) package in R was used to identify isoform expression modules within the data set and to create dendrograms and heatmaps [63].

### 4.7. Functional Annotation of Isoforms

All the identified isoforms were annotated by using the National Center for Biotechnology Information (NCBI) non-redundant protein (Nr), NCBI non-redundant nucleotide (Nt), Swiss-Prot, Protein Family (Pfam), Gene Ontology (GO) and Kyoto Encyclopedia of Genes and Genomes (KEGG) databases with BLAST and a *p*-value of 10^−7^. GO enrichment analysis was performed using TBtools with Fisher’s exact test [64].

### 4.8. Isoform Expression Analysis

First-strand cDNA synthesis was conducted with approximately 2 μg of RNA using a PrimeScript^TM^ RT Reagent Kit (TaKaRa Biotech Co., Ltd., Dalian, China). qRT-PCR was conducted using SYBR Premix *Ex Taq* (TaKaRa Biotech Co., Ltd., Dalian, China) on a CFX96 Connect real-time PCR detection system (Bio-Rad Laboratories, Inc., Hercules, CA, USA) according to the manufacturer’s instructions. Isoform-specific primers were designed to distinguish between different AS events by Oligo 7 software (Appendix A). *Ubiquitin-conjugating enzyme E2* (*UBC E2*) was selected as an internal control [65]. All reactions were performed in triplicate, both technically and biologically. The ΔCT and ΔΔCT values were calculated using the 2^−^^△△^^CT^ method.

### 4.9. Metabolite Extraction and Widely Targeted Metabolite Profiling

Metabolite extraction and analysis were performed using gas chromatography-mass spectrometry (GC-MS) as described previously [66]. Leaves in different growth periods were rapidly snap-frozen in liquid nitrogen and stored at −80 °C until further processing. For each sample, 0.1 g of freeze-dried tissue from five individuals was added to extraction buffer. Then, the leaf homogenate was transferred into a 2-mL centrifuge tube. Subsequently, the filtered homogenate was vortex-oscillated for approximately 15 s, treated with ultrasound for 10 min and placed in a 20 °C refrigerator for 1 h. Then, the homogenate was vortex-oscillated at 4 °C overnight and centrifuged at 10,000 rpm for 10 min at 4 °C. Finally, 1 mL of the homogenate was filtered through a 0.22-μm organic-phase filter into a glass vial and used for further metabolite profiling. Three biological replicates were examined for each sample.

A Waters ACQUITY UPLC HSS T3 C18 column (1.8 µm, 2.1 mm*100 mm) was selected as the chromatographic column. The temperature was maintained at 40 °C, the flow velocity was 0.4 mL·min^−^^1^, and the injection volume was 2 μL. Mobile phases A and B were H_2_O and 0.04% acetic acid and acetonitrile and 0.1% formic acid, respectively. The gradient program was as follows: (1) 95:5 *v*/*v* at 0 min, (2) 5:95 *v*/*v* at 11 min, (3) 5:95 *v*/*v* at 12 min, (4) 95:5 *v*/*v* at 12.1 min, and (5) 95:5 *v*/*v* at 15.0 min.

Linear ion trap (LIT) and triple quadrupole (QQQ) scans were captured on a QQQ-LIT mass spectrometer (Q TRAP), namely, an API 6500 Q TRAP LC/MS/MS system (ABI, Waltham, MA, USA). The Q TRAP, equipped with an ESI Turbo ion-spray interface, was operated in positive ion mode and controlled by Analyst 1.6 software (AB Sciex). The electrospray ionization (ESI) source operation parameters, instrument tuning and mass calibration were as previously described [49,67]. QQQ scans were acquired as multiple reaction monitoring (MRM) experiments with the collision gas set to 5 psi. The declustering potential (DP) and collision energy (CE) for individual MRM transitions were determined by further DP and CE optimization. A specific set of MRM transitions was monitored for each period according to the metabolites eluted within this period [68]. ANOVA was applied to the absolute metabolite quantifications, and pair-wise comparisons were performed using Tukey’s honest significance difference (HSD) test in PAST v.3.x [69]. The WGCNA (v1.42) package in R was used for metabolic correlation network analysis reference to previous studies which also used this method for metabolic correlation analysis [63,67,70]. The modules were obtained using the automatic network construction function blockwise modules with default settings.

### 4.10. Determination of CGA, Rutin and Total Flavonoid Levels

Approximately 0.5 g of freeze-dried *E. ulmoides* leaf powder was added to a test tube with approximately 20 mL of 60% ethanol. The tube was allowed to stand for 1 h and then fastened to an ultrasonic cleaner for 0.5 h. The liquid supernatant was taken in a volumetric flask (50 mL), and approximately 20 mL of ethanol (60%, *v*/*v*) was added to the residue. Then, the tube was fastened to an ultrasonic cleaner for 0.5 h for re-extraction. The liquid supernatant was transferred into a volumetric flask (50 mL), and 60% ethanol was added to obtain a final volume of 50 mL. Finally, 1 mL of liquid supernatant was extracted and further transferred into the sample bottle after filtration through a 0.22-μm microfiltration membrane.

The separation was performed using an Agilent Technologies high-performance liquid chromatography (HPLC) 1260 system (Germany) coupled to an Agilent Zorbax SB-C18 column with a length of 250 mm, diameter of 4.6 mm and particle size of 5 μm. The column temperature was maintained at 30 °C, and the injection volume was 10 μL. The mobile phase consisted of an acetonitrile solution of formic acid 0.1% (*v*/*v*) and an aqueous solution of formic acid 0.1% (*v*/*v*). The linear gradient started with a 10% acetonitrile solution, increasing to 15% in 15 min, 18% in 30 min, and 10% in 45 min. Detection was performed at 320 nm and 360 nm. HP-ChemStation software was used for data analysis (using peak area values), and the concentrations (mg·g^−^^1^) of CGA and rutin in the samples were determined using external calibration. Four biological replicates were examined for each sample.

Total flavonoids extraction was performed as described previously [71]. Semi-quantifications of flavonoids were carried out using linear regression method by comparing to the signal of rutin at 360 nm. Flavonoids accumulations were calculated in milligrams per gram of fresh weight (mg/g). Four biological replicates were examined for each sample.

### 4.11. Integrative Analysis of the Metabolome and Transcriptome

To analyse the interactions among isoform expression and metabolite concentrations associated with phenylalanine, flavone and flavonol biosynthesis, an interaction network was constructed based on Pearson correlation coefficients (PCCs). The Pearson correlation coefficient was calculated in the R environment (https://www.r-project.org/, last accessed date: 15 July 2019) with its “base” function and “stat” packages. Correlations with a coefficient of R ≥ 0.95 or R ≤ −0.95 were selected. The co-expressed isoforms with strong interconnection were considered as hub isoforms. Metabolomic and transcriptomic relationships as well as transcription factor (TF) and target relationships were visualized using Cytoscape (v.3.7.0).

### 4.12. Semiquantitative RT-PCR Verification

The selected AS events were validated by SqRT-PCR using a set of primers designed in accordance with each AS event. Total RNAs were extracted using TRIZOL solution. Next, they were treated with DNAase I (Thermo Scientific, Waltham, MA USA) and reverse transcribed to cDNA (random priming) using M-MLVRT (Promega, Madison, WI, USA). After PCR amplification and gel purification, PCR products were inserted into the pGEM-T Easy Vector System (Promega), and transferred into an *Escherichia coli* DH5α bacterial strain. Positive clones were selected through a blue-white selection experiment, and sequenced using the chain-termination method on ABI 3100 automated sequencer (ABI, Waltham, MA USA).

### 4.13. Availability of Supporting Data

We deposited the RNA-seq data in the Sequence Read Archive (SRA) database with an accession number SRP218063.

## 5. Conclusions

In conclusion, 2880 novel loci which were not annotated in *Eucommia ulmoides* genome database were identified in *E. ulmoides* based on single-molecule sequencing. A total of 515 analytes were identified from four *E. ulmoides* leaf samples using UPLC-MS. Different secondary metabolites showed different accumulation preferences during leaf growth and development. We explored the regulatory network associated with flavonoid and phenylpropanoid biosynthesis using integrated analysis of the metabolome and transcriptome and found several *MYB* and *bZIP* isoforms at the centre of the regulatory network in the biosynthesis pathway. Our findings could be used by researchers to apply bioinformatics approaches in order to elucidate the mechanism of phenylpropanoid and flavonoid regulation during *E. ulmoides* leaf growth and development and highlight the usefulness of an integrated approach for understanding this process. In addition, our findings will facilitate more accurate material selection in the forestry, health and pharmaceutical industries.

## Figures and Tables

**Figure 1 ijms-20-04030-f001:**
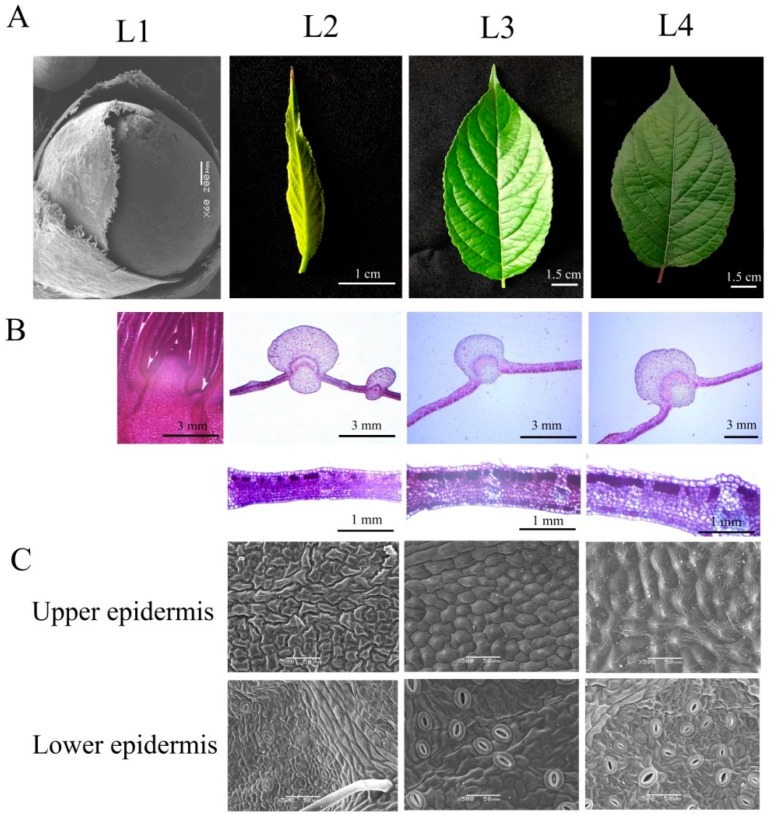
Morphological characterization of *Eucommia ulmoides* leaves at different growth stages. (**A**) Developmental process of *E. ulmoides* leaves. (**B**) Cross-sections of leaf buds and leaves. (**C**) Morphological characterization of the leaf epidermis.

**Figure 2 ijms-20-04030-f002:**
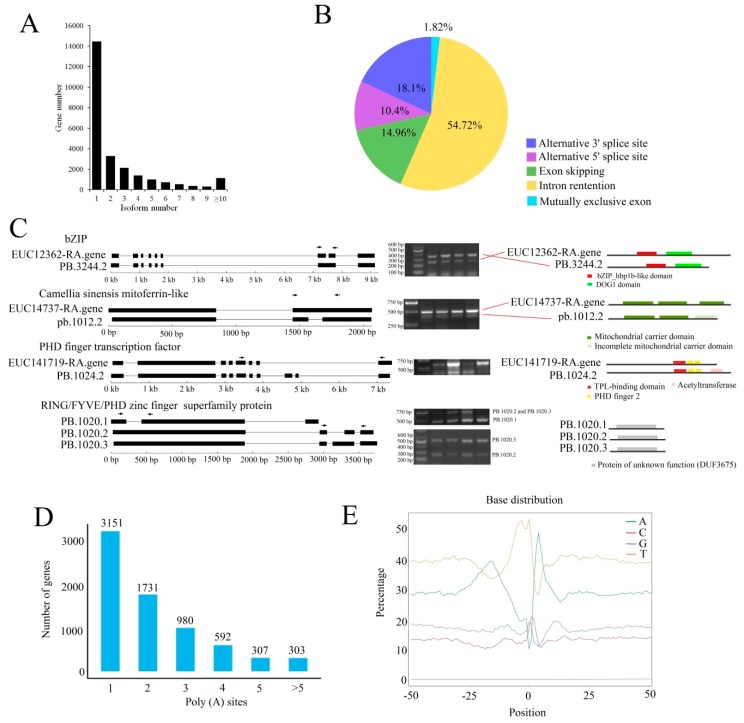
Characterization of *Eucommia ulmoides* leaf transcriptomes using isoform sequencing and second-generation sequencing (SGS). (**A**) Distribution of the number of isoforms per gene. (**B**) Distribution of different type of alternative splicing (AS) events in *E. ulmoides*. (**C**) PCR (Polymerase Chain Reaction) validation of alternative splicing events identified by Iso-Seq. The isoform model for each genes were shown on left panel. Gene models obtained from *E. ulmoides* genome database were shown on top and the novel isoforms that are supported by PacBio reads are shown under the gene. Black boxes represent for exons, lines represent introns, and arrows indicate primer sets (F, forward and R, reverse). Middle panel presented the validation of differential AS (Alternative splicing) events using RT-PCR (Reverse Transcription-Polymerase Chain Reaction). Predicted protein sequence for each isoform and conserved domains predicted using the CD-search program (NCBI, National Center for Biotechnology Information) are presented in the right panel. (**D**) Distribution of the poly (A) site per gene. (**E**) Relative frequency of each nucleotide around poly(A) cleavage sites. Sequences in the upstream (−50 bp) and downstream (+50 bp) of each poly(A) cleavage site were analysed.

**Figure 3 ijms-20-04030-f003:**
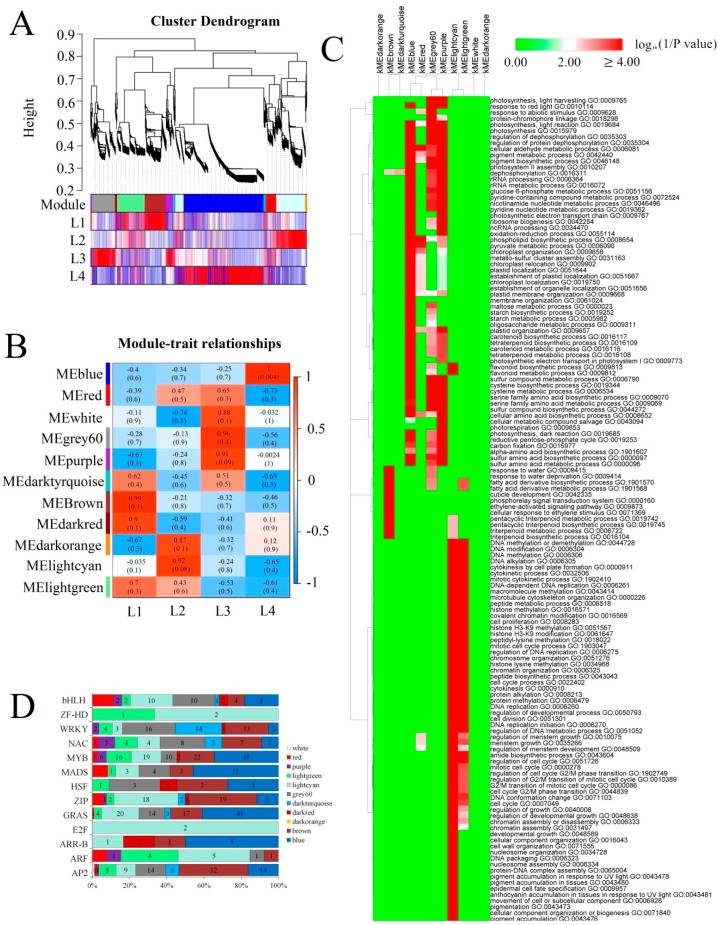
Weighted correlation network analysis (WGCNA) of all expressed *Eucommia ulmoides* isoforms. (**A**) Cluster dendrogram of expressed isoforms based on expression levels in the four growth stages. Each branch in the tree represents one isoform, and different colours represent different isoform co-expression modules. (**B**) Module–group association. The expression patterns of eleven major modules are shown by the heat map. Each row represents an expression module, while each column corresponds to a sample group. The colour of each cell at the row-column intersection indicates the correlation coefficient between the module and group. The colour bar indicates the degree of correlation from low (blue) to high (red). (**C**) Gene ontology (GO) enrichment analysis of isoforms in each cluster. The −log^10^
*p* values and adjusted *p* values ≤0.0005 are included to indicate significant enrichment. Green indicates high *p* values, and red indicates low *p* values. (**D**) Distribution of transcription factors in different modules.

**Figure 4 ijms-20-04030-f004:**
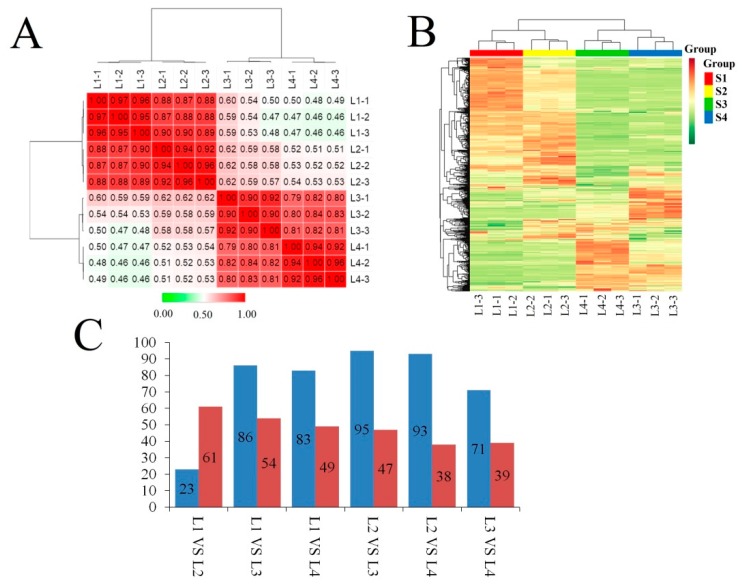
Characterization of metabolite profiles. (**A**) Correlation of metabolite accumulation between different samples. (**B**) Heat map and cluster analysis of different *Eucommia ulmoides* leaf samples at the metabolome level. Green indicates low abundance, and red indicates high abundance. (**C**) Number of differentially accumulated metabolites. The comparison of the amount of the differentially accumulated metabolites between L1 and L2; L1 and L3; L1 and L4; L2 and L3; L2 and L4; and L3 and L4 are summarized. The blue column represented for quantity of metabolites which showed higher accumulation level in former one of pairwise comparison, while red column represented for quantity of metabolites which showed higher accumulation level in latter one.

**Figure 5 ijms-20-04030-f005:**
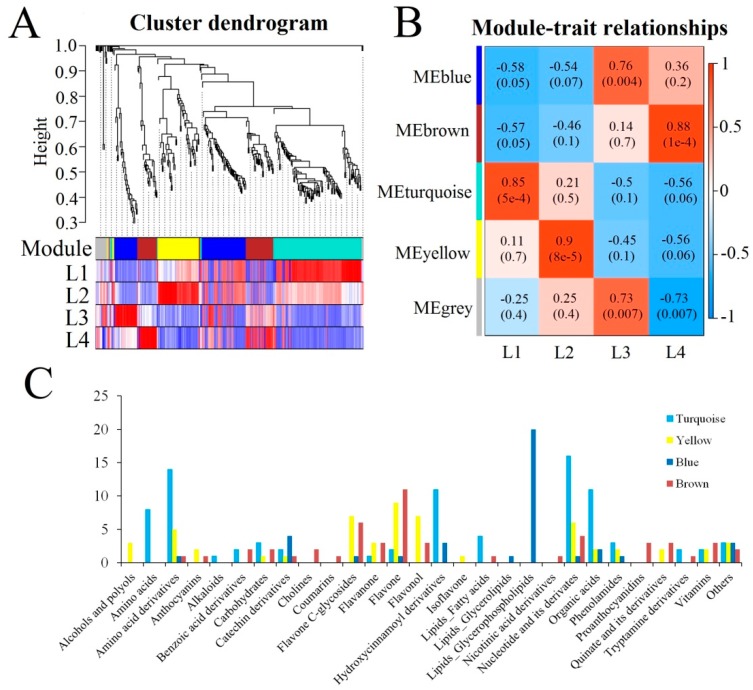
Weighted correlation network analysis (WGCNA) of all identified metabolites. (**A**) Cluster dendrogram of metabolites based on metabolome levels in the four growth stages. Each branch in the tree represents one metabolite, and different colours represent different metabolome co-expression modules. (**B**) Module–group association. Each row represents an expression module, while each column corresponds to a sample group. The colour of each cell at the row-column intersection indicates the correlation coefficient between the module and group. The colour bar indicates the degree of correlation from low (blue) to high (red). (**C**) Distribution of metabolites in different WGCNA modules.

**Figure 6 ijms-20-04030-f006:**
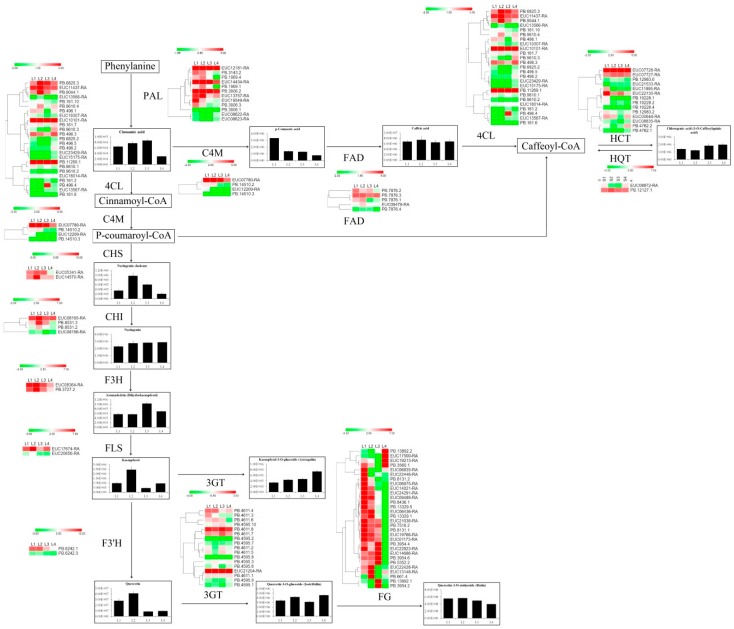
The principal pathways of flavonoid and phenylalanine biosynthesis during *Eucommia ulmoides* growth and development. These pathways were constructed based on Kyoto Encyclopedia of Genes and Genomes (KEGG) pathways and literary references. The words in the black box represent metabolites, and the metabolite levels are shown as histograms. Many macromolecular compounds, including cinnamoyl-CoA, p-coumaroyl-CoA and caffeoyl-CoA, were not detected by metabolomes. Arrows represent enzyme reactions; enzymes are shown next to the arrows. Heat maps next to the enzyme names indicate the expression value of isoforms across tissues (L1, L2, L3, and L4). The colour scale represents log2-transformed fragments per kilobase of transcript per million mapped reads (FPKM) values. Green indicates low abundance, and red indicates high abundance. The isoforms with an FPKM value less than 0.1 in all samples are not included in the heat map.

**Figure 7 ijms-20-04030-f007:**
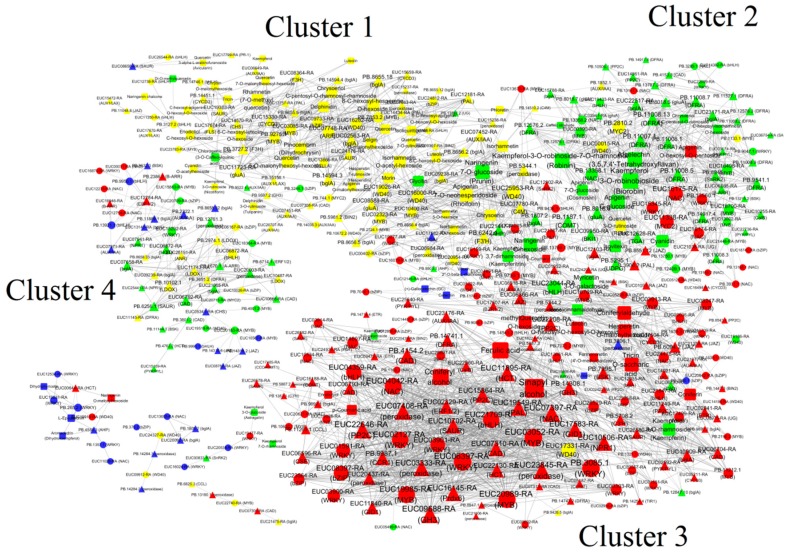
Interaction network depicting isoform–isoform interactions and relationships between regulatory isoforms and phenylalanine- and flavonoid-related metabolites. Edges or connections represent co-expression between isoforms with a Pearson correlation coefficient (PCC) < −0.95 (solid line with a T-type arrow) or ≥ +0.95 (solid line with a normal arrow). Red, yellow, blue and green represent the isoforms with the highest expression levels in leaf buds (L1), growing leaves (3 cm long, L2), young leaves (L3) and old leaves (L4), respectively. The round rectangle, triangle and ellipse represent the metabolites, functional genes and transcription factors, respectively. Larger edges in the network indicate isoform edges with more connections.

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
