# Peer review of "Dynamic Changes in Metabolite Accumulation and the Transcriptome during Leaf Growth and Development in Eucommia ulmoides"

_ijms, 2019, doi:10.3390/ijms20164030_

Round 1

Reviewer 1 Report

This manuscript reports transcritome and metabolome analyses of Eucommia ulmoides. Comprehensive data provided from this manuscript will be helpful for the researcher in the field. However, there are some concerns should be addressed before publication.

Figures in this manuscript are too small to read. I can not understand Figure 2C because the letter is not  readable. GO categories in Figure 3C cannot be read. All figures should be reconsidered before publication. Is transcriptome data deposited to public database such as GEO in NCBI? The raw data should be published.

Author Response

(1)  Figures in this manuscript are too small to read. I can not understand Figure 2C because the letter is not readable. GO categories in Figure 3C cannot be read. All figures should be reconsidered before publication.

Answer: Thank you for your suggestion. We redraw all figures with larger font size according to your suggestion. Besides we added a short figure legend for each panel of figure 2C in order to o help readers understand the figure 2C better (line 182). The revised figure legend was shown as below:    Figure 2C: PCR validation of alternative splicing events identified by Iso-Seq. The isoform model for each gene was shown on left panel. Gene models which obtained from E. ulmoides genome database were shown on top and the novel isoforms that are supported by PacBio reads are shown under the gene. Black boxes represent for exons, lines represent introns, and arrows indicate primer sets (F, forward and R, reverse). Middle panel presented the validation of differential AS events using RT-PCR. Predicted protein sequence for each isoform and conserved domains predicted using the CD-search program (NCBI) are presented in the right panel. The revisions were highlighted by Track Changes functions in figure 2C legend. For figure 3C, we used a relatively loose condition (adjusted P values ≤ 0.001) for GO enrichment analysis in previous version. This strategy lead to a large number of significant enriched GO categories was identified and retained in heatmap which resulted in a low pixel image of heatmap. In current version, we used a more stringent screening condition. Only adjusted P values ≤0.0005 are included to indicate significant enrichment. However, these changes will not result in the missing of important isoforms and important GO categories that identified in previous version. 

(2) Is transcriptome data deposited to public database such as GEO in NCBI? The raw data should be published.

Answer: Thank you for your reminding. The transcriptome data have been deposited to NCBI according to your suggestion. RNA-seq raw sequence data for the 4 samples with three biological repeats for each from this article can be found in the NCBI Short Read Archive database under the accession numbers: SRP218063. Besides the accession numbers have been supplemented at the end of revised manuscript (line 704).

Reviewer 2 Report

The authors describe a robust dataset comprising transcriptomes and metobolomes of Eucommia ulmoides. I believe it is a potentially interesting report, however in the current form there are some serious problems with the analytical strategy applied by the authors. My main concern is that they used exactly the same methodology to analyze differences in gene expression and differences in the accumulation of metabolites. There is no justification for that. Fig. 5 legend says "Weighted gene co-expression analysis (WGCNA) of all identified metabolites", see also lines 379-380. Genes are certainly not metabolites... It is incorrect to refer to metabolites as "up- or down-regulated" or "expressed" (Fig. 4c). Please, provide justification of that kind of analysis (either a well-supported theoretical framework or extensive citations to published papers using that approach). In the section "2.11. Integrative analysis of the metabolome and transcriptome" no details are given whatsoever, except for the statement that it was based on Pearson correlation coefficients. Please, provide information on the bioinformatic tools, were they publicly available or developed in-house for the particular study. Modules in Fig. 5 (metobolome co-expression [???] modules) do not correspond with clusters, why?

Minor issues:

Introduction, lines 55-56, "the molecular mechanism involved in flavonoid biosynthesis in E. ulmoides remains unknown". To me the statement is too general, as the flavonoid systhesis pathway is largely known in plants, including E. ulmoides. Obviously, regulatory mechanisms should be elucidated on the species level.

Methods, line 100, "Annual" refers to a plant that completes the vegetative cycle in one year. I believe the authors meant "one-year-old".

Methods, lines 14-140, there is a self-contradictory statement "Only FLNC ROIs were retained..." and then "The FLNC ROIs were clustered by iterative clustering for error correction (ICE) using SMRT Analysis (v2.3.0) software to generate the consensus clusters of all FLNC, non-FL, and chimeric sequences" - weren't non-FL and chimeric sequences removed at earlier stages?

Why conclusions are given in the last par. of the section 4.3? they should be separate.

Why in the legends of Tables S5 and S6 moso bamboo (Phyllostachys edulis) is mentioned?

In summary, I believe the paper is of interest to IJMS readers, provided that the metabolome data and transcriptome/metabolome relationships are adequately analyzed.

Author Response

The authors describe a robust dataset comprising transcriptomes and metobolomes of Eucommia ulmoides. I believe it is a potentially interesting report, however in the current form there are some serious problems with the analytical strategy applied by the authors. My main concern is that they used exactly the same methodology to analyze differences in gene expression and differences in the accumulation of metabolites. There is no justification for that. Fig. 5 legend says "Weighted gene co-expression analysis (WGCNA) of all identified metabolites", see also lines 379-380.

Answer: Thank you for your reminding. The description of ‘Weighted gene co-expression analysis (WGCNA) of all identified metabolites’ was incorrect, and we changed the description into ‘Weighted co-expression analysis (WGCNA) of all identified metabolites’ according to previous studies (Wang et al., 2013; Wang et al., 2014) (line 276 and line 286). The studies of Wang et al. 2008 and Wang et al. 2013 also used this method to reveal the metabolic modules.

Reference:

Wang JX, Chen L, Tian XX, Gao LJ, Niu XF, Shi ML, Zhang WW. Global metabolomic and network analysis of Escherichia coli responses to exogenous biofuels. Journal of Proteome Research (2013) 12: 5302−5312.

Wang JG, Zhang XQ, Shi ML, Gao LJ, Niu XF, Te RG, Chen L, Zhang WW. Metabolomic analysis of the salt-sensitive mutants reveals changes in amino acid and fatty acid composition important to long-term salt stress in Synechocystis sp. PCC 6803. Functional & Integrative Genomics (2014) 14: 431-440.

Genes are certainly not metabolites. It is incorrect to refer to metabolites as "up- or down-regulated" or "expressed" (Fig. 4c). Please, provide justification of that kind of analysis (either a well-supported theoretical framework or extensive citations to published papers using that approach).

Answer: We revised our description refer to accumulation of metabolites according to your suggestion. The words ‘The number of differentially expressed metabolites’ changed into ‘The number of differentially accumulated metabolites’ in current version (Line 261). Besides, the legend of figure 4C was always revised and shown as below (Line 270):

Number of differentially accumulated metabolites. The comparison of the amount of the metabolites between L1 and L2; L1 and L3; L1 and L4; L2 and L3; L2 and L4; and L3 and L4 are summarized. The blue column represented for quantity of metabolites which showed higher accumulation level in former one of pairwise comparison, while red column represented for quantity of metabolites which showed higher accumulation level in latter one.

 Besides, we cited three published papers that our work refer to in section of ‘4.9. Metabolite extraction and widely targeted metabolite profiling’. Among these, the studies of Wang et al. 2008 and Wang et al. 2013 also used the WGCNA method to reveal the metabolic modules.

Reference:

Langfelder P, Horvath S. WGCNA: an R package for weighted correlation network analysis. BMC Bioinformatics (2008) 9: 559.

Wang JX, Chen L, Tian XX, Gao LJ, Niu XF, Shi ML, ZhangWW. Global metabolomic and network analysis of Escherichia coli responses to exogenous biofuels. Journal of Proteome Research (2013) 12: 5302−5312.

Wang JG, Zhang XQ, Shi ML, Gao LJ, Niu XF, Te RG, Chen L, Zhang WW. Metabolomic analysis of the salt-sensitive mutants reveals changes in amino acid and fatty acid composition important to long-term salt stress in Synechocystis sp. PCC 6803. Functional & Integrative Genomics (2014) 14: 431-440.

In the section "2.11. Integrative analysis of the metabolome and transcriptome" no details are given whatsoever, except for the statement that it was based on Pearson correlation coefficients. Please, provide information on the bioinformatic tools, were they publicly available or developed in-house for the particular study.

Answer: The pearson correlation coefficient in this study was calculated in the R environment (https:// www.r-project.org/) with its “base” function and “stat” packages. We provided the information of method in the section of ‘4.11. Integrative analysis of the metabolome and transcriptome’ (line 663-666).

Modules in Fig. 5 (metobolome co-expression [???] modules) do not correspond with clusters, why?

Anwser:Thank you for your reminding. We revised the sample name in Figure 5A. All samples were uniformly named as L1, L2, L3 and L4.

Minor issues:

Introduction, lines 55-56, "the molecular mechanism involved in flavonoid biosynthesis in E. ulmoides remains unknown". To me the statement is too general, as the flavonoid systhesis pathway is largely known in plants, including E. ulmoides. Obviously, regulatory mechanisms should be elucidated on the species level.

Anwser:Thank you for your suggestion, we introduced some flavonoid biosynthesis related studies in our revised manuscript (line 56).

Methods, line 100, "Annual" refers to a plant that completes the vegetative cycle in one year. I believe the authors meant "one-year-old".

Anwser: We amended this fault in our revised manuscript (line 498).

Methods, lines 14-140, there is a self-contradictory statement "Only FLNC ROIs were retained..." and then "The FLNC ROIs were clustered by iterative clustering for error correction (ICE) using SMRT Analysis (v2.3.0) software to generate the consensus clusters of all FLNC, non-FL, and chimeric sequences" - weren't non-FL and chimeric sequences removed at earlier stages?

Anwser:  Thank you for your reminding, we removed the words ‘Only FLNC ROIs were retained for downstream analysis’ in our revised manuscript (line 550).

Why conclusions are given in the last par. of the section 4.3? they should be separate.

Answer: We separated the section of Conclusions from Discussion according to your suggestion.

Why in the legends of Tables S5 and S6 moso bamboo (Phyllostachys edulis) is mentioned?

Answer: I am sorry for these basic mistakes, we revised the titles of Tables S5 and S6 in our revised manuscript.

In summary, I believe the paper is of interest to IJMS readers, provided that the metabolome data and transcriptome/metabolome relationships are adequately analyzed.

Round 2

Reviewer 2 Report

I have no further comments to the manuscript